# The Influence of Grain Legume and Tillage Strategies on $CO_2$ and $N_2O$ Gas Exchange under Varied Environmental Conditions

Emilie Marie Øst Hansen [1], Henrik Hauggaard-Nielsen [1], Eric Justes [2], Per Ambus [3,*] and Teis Nørgaard Mikkelsen [4]

[1] Department of People and Technology, Roskilde University, DK-4000 Roskilde, Denmark; emilieh@ruc.dk (E.M.Ø.H.); hnie@ruc.dk (H.H.-N.)
[2] CIRAD, Persyst Department, F-34398 Montpellier, France; eric.justes@cirad.fr
[3] Department of Geosciences and Natural Resource Management, University of Copenhagen, DK-1350 Copenhagen, Denmark
[4] Department of Environmental Engineering, Technical University of Denmark, DK-2800 Kgs. Lyngby, Denmark; temi@env.dtu.dk
* Correspondence: peam@ign.ku.dk; Tel.: +45-3533-6626

**Abstract:** By this in vitro study addressing greenhouse gas (GHG) emissions from soil-plant meso-cosms, we suggest a method to investigate the joint effects of environmental conditions, growth of plants, and agricultural soil management. Soils from two long-term agricultural trials in France were placed in climate chambers. The rotation trial was with or without grain legumes, and the tillage trial used plowing or reduced tillage. Environmental conditions consisted of two contrasting temperature regimes combined with ambient (400 ppm) or high (700 ppm) $CO_2$ concentrations in climate chambers. The plant growth went from seeding to vegetative growth. Carbon dioxide gas exchange measurements were conducted in both soil types for a period representing initial plant growth. The $CO_2$ exchange was influenced by the growing plants increasing the mesocosm respiration and gross ecosystem production. The environmental settings had no noticeable impact on the $CO_2$ exchange in the soils from the legume trial. The $CO_2$ exchange from the tillage trial soils exhibited variations induced by the environmental conditions depending on the tillage treatment. The $N_2O$ emission measurements in the legume trial soils showed little variability based on rotation, however, in soils with legumes, indications that higher temperatures will lead to more $N_2O$ emission were seen.

**Keywords:** net ecosystem exchange; gross ecosystem production; mesocosm; rotation; soil tillage

## 1. Introduction

Gas exchanges are influenced by the interactions between soils, plants, and atmospheric conditions. This paper suggests an experimental approach to include climatic conditions in the investigation of the gas exchanged from systems with both plants and soils.

Soils contain the largest carbon pool of the carbon cycle [1,2] with land use change from permanent vegetation to temporary vegetation causing the highest losses of carbon from the soil [3–6]. Soil organic carbon is an indicator of soil fertility, and a general decline in soil carbon content in agricultural soils is assumed to cause yields to stagnate [7] strongly influencing food security [8].

Agricultural soils are kept in an early stage of succession and the majority of aboveground biomass produced each year is removed. This affects both the productivity of the soil and the emissions of greenhouse gases (GHG) from the soil [9]. The pedogenetic conditions at a specific site is beyond alterations by agricultural activities, as are atmospheric $CO_2$ concentration, precipitation, solar radiation, and temperature. The farmer may on the other hand decide on activities such as soil tillage, crop choice, species and timing, irrigation, fertilization, pesticides, etc., and these decisions affect the soil structural

properties, gas exchange, and productivity [10]. No-tillage practices that conserve soil carbon are suggested as means to also reduce greenhouse gas emissions and increase soils' resilience toward the impacts of climate change [11,12]. Inclusion of legumes in cropping sequences provide an array of ecosystem services including the ability to reduce greenhouse gas emissions while accelerating rates of carbon sequestration in soil [13,14].

Microbial decomposition processes and soil biota metabolic respiration shape the soil emissions. The gases are released into the atmosphere by diffusion and mass flow, the conditions for which soil structure and atmospheric pressure regulate. This microbial outlet of gases drives the respiration from soil without a plant cover. The soil respiration based on soil organic matter (SOM) is therefore regulated by the factors regulating microbial turnover and growth [15,16]. That means the feed sources such as organic matter availability, and the C/N ratio in the soil indicate how easily the organic matter can be turned into microbial biomass and the availability of nitrogen in various forms is important for the formation of gas concentrations and emissions [17–19]. Temperature and oxygen in the soil influence microbial activity [20]. Regional climate and weather as well as soil nutrient levels and biological balance influence the $CO_2$ gas exchange of the soil [21,22]; and plant photosynthesis contributes to the soil organic matter input driving the gas exchange in the soil–plant system [19,23]. When the soil is plant-covered the respiration of the soil microbial life interacts with plant roots and plant root exudates and respiration. The plants photosynthesis emits oxygen and takes up the $CO_2$ whereas the same above ground matter emits carbon dioxide as part of its metabolic respiration [24].

The conditions for nitrogen turnover processes depend on physico-chemical parameters, soil biota, and climatic conditions [25]. Agricultural decisions on tillage influencing soil structure and meso-faunal habitats are a focal point for changing agricultural GHG emissions [26]. Compared to natural soil systems, the abundance of plant available nitrogen is high in agricultural soils, as nitrogen is essential for plant growth, but nitrogen availability is also important for soil carbon storage [27,28]. The soil production and emission of nitrous oxide is driven by the microbial processes that reduce nitrate elemental-free dinitrogen interrupted by a partial availability of oxygen [29].

With the simulations of different future-climate spring conditions in climate chambers the present included $2 \times 2$ sets of intact soil columns taken in long-term agricultural experiments. Soil from northern France that had been plowed or not plowed leads to changes of organic matter content and oxygen availability in the soil [30]. Another soil from southern France that had been cultivated with different levels of legumes in the crop rotation [31] had changing nutrient availability and thereby microbial activity and plant nutrition [32]. Influenced by the agricultural practices and climate factors this study provides one overarching expression for the processes influencing the gas exchanges when measuring the $CO_2$ gas exchange from the soil–plant ecosystem. We use the common concepts of ecosystem respiration (ER) and net ecosystem exchange (NEE) [33] in free-air carbon enrichment studies [34]. The ER measured as the flux of $CO_2$ from the soil, expresses the metabolic activity in one specific ecosystem with its specific history and under the prevailing conditions. NEE encompass the $CO_2$ emission by microbial and plant metabolic respiration simultaneously with photosynthetic assimilation of $CO_2$. The relation between photosynthetic and metabolic activities leads to $CO_2$ gas exchange results that may be close to zero if photosynthesis and respiration are of the same size; have a negative value if photosynthesis drives the gas exchange and; a positive value if respiration drives it. To estimate the soil–plant ecosystem's carbon uptake, the ER is subtracted from the NEE, to yield an expression for the gross ecosystem production (GEP). The GEP is represented by negative values; a carbon gain for plants is a carbon decrease in the atmosphere.

Focusing on soil tillage and crop rotation the aim of the present study was to suggest an approach to investigate how these agricultural practices influence the gas exchange of intact soil columns sampled from long-term field experiments when exposed to changes in temperature and $CO_2$ levels.

## 2. Materials and Methods

### 2.1. Soils

Two French soils were sampled for use in this study. One from Auzeville-Tolosane, (southern France; 43°31′39.97″ N 1°28′56.64″ E) and one from Estrée-Mons (northern France; 50°38′31.22″ N 3°06′27.62″ E). The soils were sampled as intact soil columns in PVC pipes of height 45 cm and diameters 14.7 (Auzeville-Tolosane) and 15.5 cm (Mons). The sampling was executed by pressing the cut lengths of PVC pipe into the ground with an excavator shovel. Soil characteristics from the long-term experiments are shown in Table 1.

**Table 1.** The particle size distribution, organic carbon, and total nitrogen content of the soils used in this study.

| Texture/Site | Auzeville-Tolosane [1] | | Mons [2] | | |
|---|---|---|---|---|---|
| | GL1-CC | GL0-BF | Before Exp. | RT | CONV |
| Clay, % | 26 | 23.7 | | 18.7 | |
| Silt, % | 34.2 | 33 | | 74.4 | |
| Sand, % | 35.5 | 41.5 | | 6.9 | |
| Organic C, g C kg$^{-1}$ | 8.9 | 7.4 | 10.4 | $12.0 \pm 0.1$ | $9.8 \pm 0.1$ |
| Total N, g N kg$^{-1}$ | 1.05 | 0.82 | 1.0 | $1.4 \pm 0.0$ | $1.3 \pm 0.0$ |

[1] The soil textures of the Auzeville-Tolosane sites were determined in 2003 before the rotation experiment started, and the carbon and nitrogen contents were determined in 2012 [35]. The Auzeville-Tolosane rotation treatments used in the present experiment are the one grain legume + cover crops (GL1-CC) and no grain legumes + bare fallow (GL0-BF). [2] The Mons soil particle size distribution was determined in 2006 (before exp.) and the long-term tillage experiment initiated in 2010 [31]. Later the differences in carbon and nitrogen resulting from the experiment were determined [36]. The Mons soil treatments used in the present experiment are the reduced tillage (RT) and plowed (CONV) treatments.

A note on word use: the soils were sampled as "intact soil columns," however when those samples were handled in the experiment it was as "mesocosms" (they were no longer simply soil because plants were added, they were no longer "intact" because they were equipped with experimental gear). "Soil sample(s)" may be used to refer to the individual PVC pipes with soil that were the experiment's mesocosms.

Auzeville-Tolosane soil had been exposed to one of two different rotations in an experiment established in 2003: One grain legume + cover crops (GL1-CC) or no grain legumes + bare fallow (GL0-BF). Further experimental details are described in Plaza-Bonilla et al. [31].

The Mons soil had been exposed to one of two different tillage strategies since 2010: Plowed (CONV) or reduced tillage (RT). After five years of the tillage experiment a significantly higher level of total carbon was found in the reduced tillage soil [36]. Further details of the tillage experiment are described in Coudrain et al. [30].

### 2.2. Experimental Design

To study the soil management's influence on the combined soil-plant ecosystem, we strived to equalize the input to what the columns would provide depending on their management history: the hydraulic properties or nutrient composition may have been altered by the soil management decisions, we wanted it to be reflected in the set-up. The nutrient differences could be reflected in the experiment by not adding additional fertilizer to the columns, so that a benefit from having grain legumes in the rotation would be accounted for by the growing plant. The potential influence from soil management on hydraulic properties could be reflected by allowing the columns to provide water for the plants depending on the soils hydraulic properties. By connecting the columns' bottoms to capillary mats and wicks and to water beneath the columns, the water accessibility was in principle the same; however, the soil would provide water for the plants depending on the structure allowed by soil texture and structural alterations arising from management differences.

The bottoms of the intact soil columns were each equipped with a plastic end cap (from DBIplastics, Stenlille, Denmark) with a hole (Figure 1, detail). Inside the end cap two layers of cotton "gardener's fabric" were placed as capillary matting; the one touching the soil was joined by stables to two 70 cm long, 7 mm wide cotton wicks (from Fakkelfabrikken, Assens, Denmark) that went through the second layer and the end cap. Between the layers of capillary fabric, plastic pellets $1 \times 2$ mm were placed to ensure the hydraulic contact between the soil bottom and the upper capillary mat.

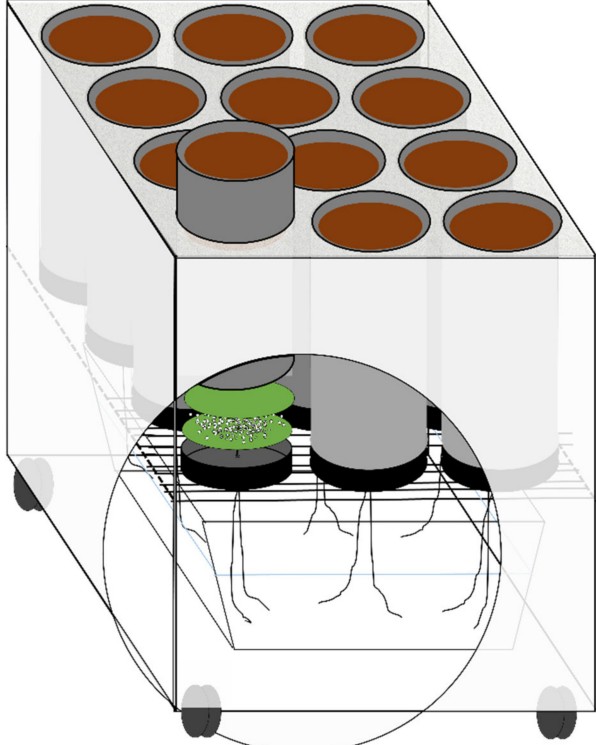

**Figure 1.** The modified freezer.

The soil samples were placed in four modified chest freezers (Figure 1) where temperature was adjusted by thermostats (Universaltermostat UT-200) to keep soil temperature at a level between day and night air temperatures. The chest freezers (Scandomestic SB 451) with an internal volume of 368 L were modified to contain 12 soil samples; three samples of the two soil types and their two management treatments.

For each freezer the lid was removed, a wooden rack was built to keep the soil surfaces aligned with the freezer opening, a fan (SUNON Axial Fan) was supplied to mix the freezer air, and a layer of 2 cm thick double reflective insulation on top to keep the cool air in the freezer. The wicks from the soil columns were placed into a water-filled plastic case underneath the rack in the freezer to provide water for the samples through the wicks. The freezers were equipped with wheels.

The progress of the experiment ensured that the soil–plant mesocosms went through various stages of spring interactions before the vegetation grew too dense and root growth expectedly too restricted. Initially when the soil was bare, barley was sown, and plants emerged and grew. After 23 days the plants were partially harvested ("grazed"), and exposed to a 10-day long increase in temperature. At all stages, the $CO_2$ concentration changes were measured. During the plant growth period, $N_2O$ gas was sampled from soils of rotation-managements in A and CT climate treatments.

### 2.3. Plants

In each of the 48 soil samples, eight barley seeds were sown (*Hordeum Vulgare* L., cultivar Evergreen). The plants were watered with 0.1, 0.15, or 0.2 L tap water every other day depending on plant development, to ensure plant growth regardless of the water availability provided by the soil properties. The plants were sown on 27th May and fully harvested after 34 days of growth on 30th June.

### 2.4. Climate Treatments

The experiment was conducted in the RERAF climate chambers that provide a controlled environment and uniform conditions, thus eliminating other potentially interacting parameters. The RERAF phytotron (Risø Environmental Risk Assessment Facility, Technical University of Denmark, Risø, Denmark) consists of six gastight chambers sized $6 \times 4 \times 3$ m, providing control of temperature, $[CO_2]$, air humidity, and light. In this study, four chambers were used and treatment variables were daily controlled to agree with the set points using graphics with set points and measurements graphs. Detailed descriptions of RERAF can be found in e.g., Ingvordsen et al. [37].

The experiment contained a period of initial plant growth, and the climate treatments then resembled spring growth conditions, which expectedly will change substantially over the course of this century because of changes in global climate. The selected treatments (Figure 2) were four combinations of two levels of temperature and two concentrations of $CO_2$. The low air temperature level (A) resembles temperatures that are common at the geographical origin of the soils; representing at 17 °C/10 °C day/night an average month of May in Mons, and an average month of April in Auzeville-Tolosane (Table 2).

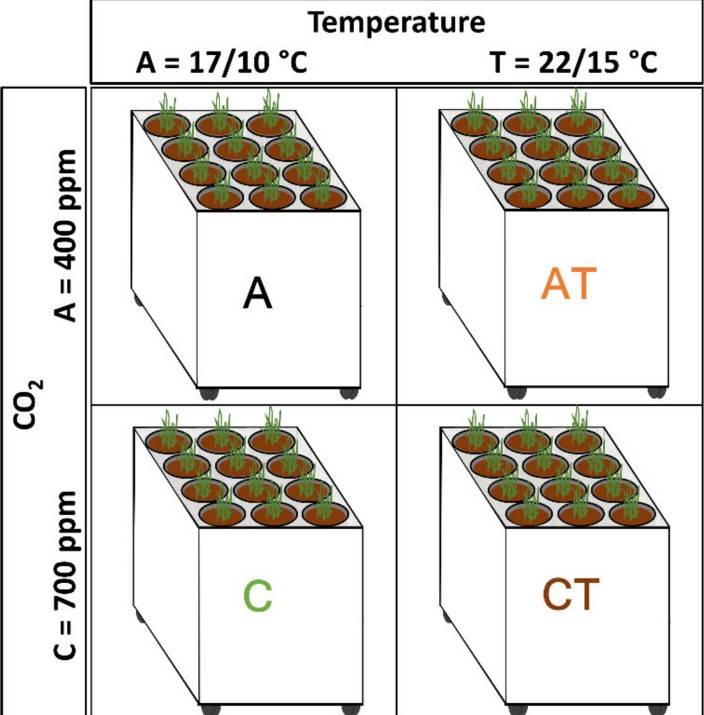

**Figure 2.** The climate treatments' compositions of temperature and $CO_2$ concentration with the low (A and C) and high (AT and CT) temperatures resembling the common values at the geographical origin of the soils influenced by ambient (A) and increased (C) atmospheric $CO_2$-concentrations, respectively.

**Table 2.** High and low average monthly air temperatures (°C) of the soils' geographical origins.

|  | Mons | | Auzeville-Tolosane | |
| --- | --- | --- | --- | --- |
|  | **Low** | **High** | **Low** | **High** |
| January | 0.6 | 5.5 | 2.4 | 9.5 |
| February | 0.6 | 6.6 | 3 | 11.1 |
| March | 3 | 10.6 | 5 | 14.5 |
| April | 4.5 | 14 | 7.1 | 17 |
| May | 8.2 | 17.9 | 10.9 | 21 |
| June | 10.6 | 20.7 | 14.3 | 25.2 |
| July | 12.5 | 23.4 | 16.5 | 28 |
| August | 12.4 | 23.4 | 16.5 | 27.9 |
| September | 10.1 | 19.6 | 13.4 | 24.6 |
| October | 7.3 | 14.9 | 10.5 | 19.5 |
| November | 3.6 | 9.3 | 5.8 | 13.3 |
| December | 1.3 | 5.9 | 3.2 | 9.9 |

Data from MétéoFrance.com, accessed on 4 November 2019 [38].

In contrast to A, the high temperature level (AT) provides 22 °C/15 °C day/night and thus much warmer spring conditions. The high temperature level can be found as average July or August temperatures in Mons, and average May or June temperatures in Auzeville-Tolosane. The low level of $[CO_2]$ was set to mimic current atmospheric concentration of 400 ppm and the high level (C) was set to resemble a concentration expected at the end of the 21st century of 700 ppm [39].

Throughout the experiment, the set points for the thermostats in the freezers were 12 °C and 17 °C in the colder and warmer treatments respectively. Although the control of all treatment variables worked satisfactorily, the freezers with the mesocosms were moved into a new chamber every week to equalize any unknown effects caused by the individual chambers.

### 2.5. Measurements of Gas Concentrations: $CO_2$, $N_2O$

Gas concentration measurements took place in the climate treatments under light or dark conditions; NEE was measured in daytime conditions with an LED lamp placed on top of the measuring chamber facing downwards resulting in 457–467 μmol m$^{-2}$ s$^{-1}$ photosynthetically active radiation (PAR). ER was measured under darkened conditions where the chamber was covered by a costumed-fitted, double-reflective insulation top that reduced light to <0.5 μmol m$^{-2}$ s$^{-1}$ (i.e., to within the uncertainty of the light meter).

Measurements were done in a closed flux system that was moved from mesocosm to mesocosm. The measuring chamber was built of clear Plexiglas, with an inner diameter of 15 cm and height 30 cm (Figure 3). The bottom had a 4 cm flat collar under which soft, self-rising material was attached to provide airtightness between the chamber and the PVC pipes. Inside the chamber, a fan provided air mixing [40]. Two Teflon tubes connected the chamber to a LI-6262 $CO_2$/$H_2O$ Analyzer (LI-COR, Lincoln, NE, USA), which measured the concentration of $CO_2$ in the chamber continuously hence providing information on $CO_2$ concentration increase or decrease.

Gas samples were taken from the headspace of the measuring chamber of Auzeville-Tolosane soil mesocosms with the purpose of detecting whether the rotational differences regarding grain legumes could influence the $N_2O$ emissions from the agricultural soils under different climatic settings. Gas sampling was done using a manual sampling technique, where samples were taken from the headspace of the chamber at 0, 15, 30, 45, 60 min using a 50 mL syringe and pre-evacuated glass Exetainers (Labco., High Wycombe, UK). The concentrations of $N_2O$ were determined at University of Copenhagen by the use of Gas Chromatography (HP7890A, Agilent, Wilmington, NC, USA).

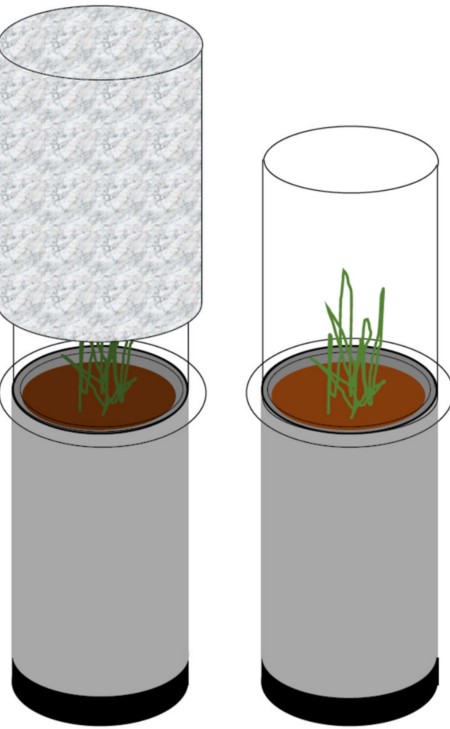

**Figure 3.** Soil columns and measuring chamber, with and without darkening hood. Chamber height 30 cm, diameter 15 cm, soil column height 45 cm, inner diameter 15 cm.

For both $CO_2$ and $N_2O$ the measured concentrations were converted into fluxes. The rate of change (the slope of linear regression) relative to the sampling area (the soil column area) was converted to fluxes, e.g., µmol $CO_2$-C $m^{-2}$ $s^{-1}$. Linear regression was used consistently on linear parts of measurements, disregarding initial turbulence from placing the measuring chamber.

Before plant emergence, respiration was measured twice; before and after sowing. The ER and NEE were measured on four different dates, nitrous oxide was sampled on three dates, and after the partial harvest ER and NEE were measured five times with one or two days between each measurement.

### 2.6. Temperature Campaign and Partial Biomass Harvest

At 23 days after sowing, the biomass was partially harvested, leaving 0, 50, or 100% of the aboveground plant biomass in the three columns of similar origin in each climate treatment. The cut was made by a visual assessment to leave 50% biomass, the length of leaves as well as tiller fullness near pot surface resulted in biomass left at a height of approx. one-third of the length of the longest leaf. Grazing and haymaking influence the root–soil interactions and the gas exchange patterns at least on long-term grazed fields [41], and by partially harvesting the aboveground biomass we investigate the importance of plant cover during temperature increases.

The temperature campaign lasted 10 days with incremental increases of temperatures and measurements of gas exchange. Initially both temperature-levels were lowered by 5 degrees, the mesocosms left to equilibrate for two days, then $CO_2$ was measured under light and dark conditions, and the day/night/freezer temperatures were increased by 4 degrees, and the mesocosms left to equilibrate. The increase, equilibration, and measurements were repeated twice to yield a total increase in temperature of 12 degrees over the 10 days.

By rating the length in cm of the leaves to the number of days of growth, a growth rate for the two observed periods of plant growth were sketched, indicating the growth strategy of grass crops under different conditions of soil and climate.

### 2.7. Data Analyses

Data analyses on fluxes were done using R [42], MS Excel 2016, ANOVA tests, and pairwise t-tests. Analyses were done separately on data from Mons and Auzeville-Tolosane soils to investigate the impact of the treatments rather than the overall performance of one soil composition against another. When necessary to obtain homogeneity of variance, data was e.g., log-transformed based on boxcox plots in R.

## 3. Results and Discussion

The soil samples used in this experiment were collected at original agricultural field sites and transported to the experimental test facility with climatic treatments. The soil samples were all handled in a similar manner, but the measurements performed on the samples relate to the removed soils in the climate treatments, and may not be directly related to the geographical site they originated. The two soil types are different and results from the mesocosms of e.g., Mons should be compared only to the results from other Mons mesocosms.

### 3.1. Ecosystem Respiration, Net Ecosystem Exchange, and Gross Ecosystem Photosynthesis

Measurements of ecosystem respiration and net ecosystem exchange from four different dates contribute to the diagrams in Figure 4.

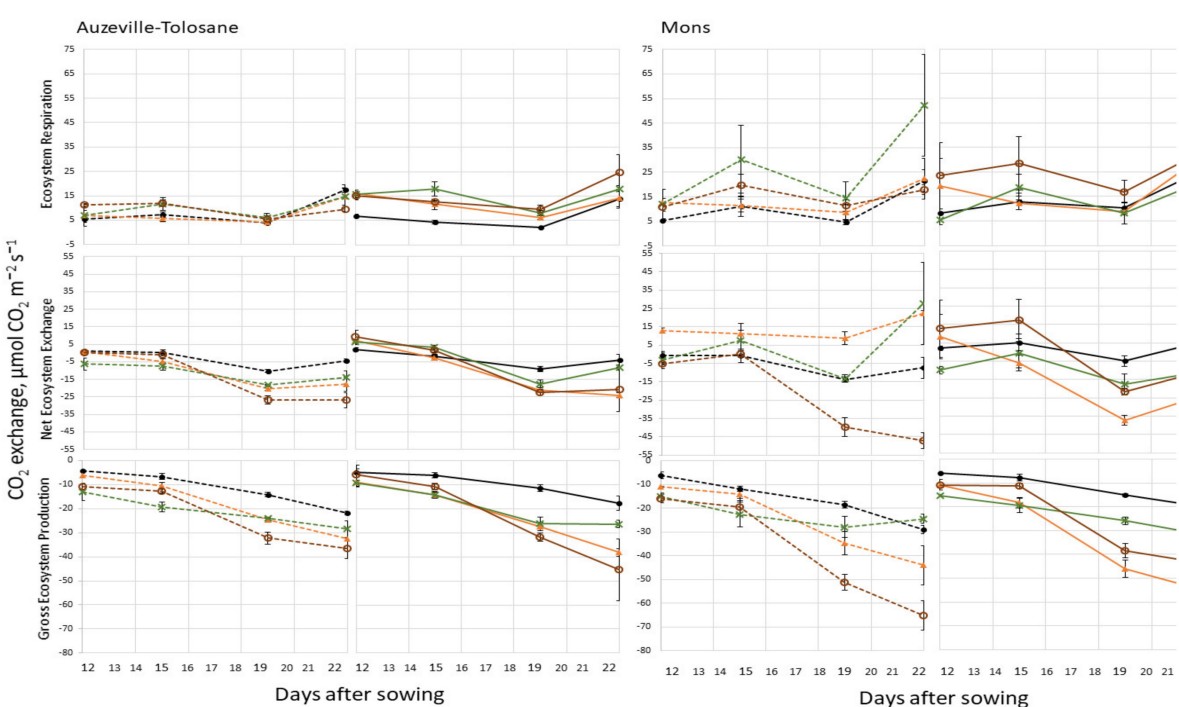

**Figure 4.** The ecosystem respiration (ER), net ecosystem exchange (NEE), and the gross ecosystem production (GEP) of the Mons (**left**) and the Auzeville-Tolosane (**right**) soils, at 2–3 weeks after sowing. The dashed lines are the control treatments (CONV and GL0-BF) soils, and the solid lines represent the experimental treatments (RT and GL1-CC) soils. The colors represent the climate treatments (for further information see Figure 2); black dots are A, orange triangles the increased temperature T, green x'es are carbon-enriched C, and open brown circles are combined carbon-enrichment and raised temperatures; CT. The error bars indicate +/− SE.

With time, across the two soil types and their treatments, a general increase in activity was found (Figure 4), which can be related to the increase in biomass that causes more ER and higher uptake of $CO_2$ (GEP). The rotation differences in the Auzeville-Tolosane soil induce no differences in ER and GEP, however there seems to be differences arising from the interactions between the soil and climate treatments. The differences in ER and

GEP between the plowed and reduced tillage Mons soils seem to stem primarily from interactions between soil and climate treatments.

In the Auzeville-Tolosane soils, the different rotations induced no significant difference in either ecosystem respiration or gross ecosystem production (Figure 4). Soil nitrogen content is important for the soil respiration as it is potentially a limiting factor for plant growth; however, the quality of an added nitrogen source determines the impact it has on the soil respiration. López-López et al. [43] detected increases in soil respiration with the addition of organic material but decreases in soil respiration with the addition of NPK fertilizer. The Auzeville-Tolosane soil ER have in all climate treatments a little higher measured values in the grain legume rotation (GL1-CC) compared to the rotation without (GL0-BF), which aligns with literature that the input of nitrogen with a carbon source increases respiration compared to an input of nitrogen without organic carbon.

In the Mons soil, the higher carbon content in the RT soil is not immediately resulting in an increased respiration compared to the CONV soil when exposed to higher temperatures, however, in a short time frame as presented here, the ecosystem respiration is composed primarily of the autotroph respiration from plants and roots [18]. While the CONV soil seems to capture more $CO_2$ in the CT climate, the difference to the RT soil is not significant. The higher carbon content in the RT soil does not induce a significantly higher respiration at higher temperatures, and the GEP at the higher temperatures is comparable to the GEP of the CONV soil. The agricultural management decision not to plow can be expected to maintain its higher levels of soil carbon compared to a plowed soil under changing climates judging from this spring setting study. The literature suggests reduced soil tillage as a means of reducing emissions of both $CO_2$ an $N_2O$ [12]. However, this study observed no significant differences between the soils of different tillage managements, but it is noteworthy that unaltered $CO_2$ emissions in combination with a higher soil carbon content are promising for an increase of soil carbon stock by reducing tillage.

### 3.2. Respiration

Respiration was measured twice on the soil samples before plant emergence; before and after sowing (Figure 5).

The respiration of the soil columns in the climate treatments was influenced by sowing resulting in more cases of significant difference after sowing than before. Carbon enrichment is known to increase soil respiration consistently [44]. From the respiration patterns of the soils in climate treatment C, it seems that soil respiration under carbon enrichment interacts with both soil type and management treatment creating more difference compared to the other climates in the plowed soil (CONV), and in the grain legume rotation (GL1-CC).

While soil respiration may reflect the water conditions in the soils [45], water content here reflects the soil management treatments as soil properties supply the mesocosms with water and is hence considered an integrated feature of the agricultural decision as well as the soil type.

Changes in soil respiration due to sowing and seed germination must be detected in changes between the same soil and climate, and there seems to be differences as respiration in most soil * climates increases yet in a few it decreases. There is e.g., a small decline in the respiration of the ATs except GL0-BF_AT. Changes in the respiration due to sowing can be caused by emergence of roots and root exudates as the seed germinates and thereby causing interactions between the plant activity and the soil microbial community.

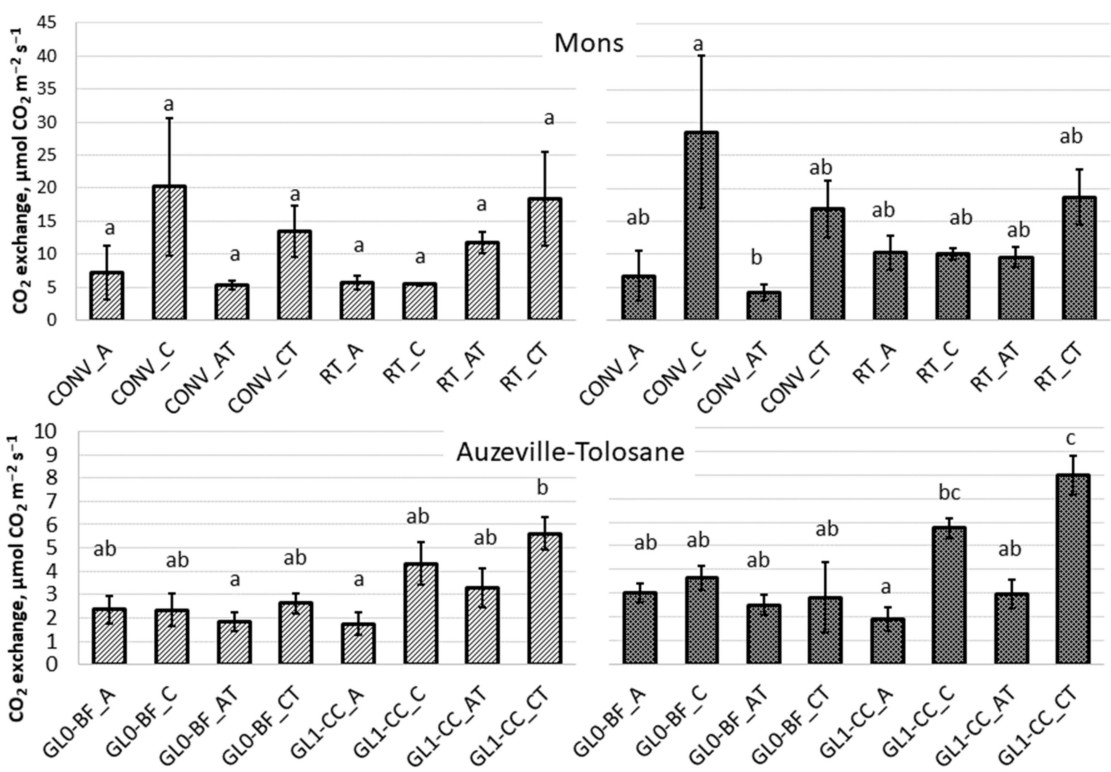

**Figure 5.** Respiration measured on the Mons (**top**) and the Auzeville-Tolosane (**bottom**) soils, without plants: light color (**left**) before sowing and dark color (**right**) after sowing. Letters indicate cases of no significant difference. Error bars indicate $+/-$ SE. For further information about x-axis treatment abbreviations see Table 1 and Figure 2.

### 3.3. Temperature Campaign and Biomass Cut

In mesocosms where all biomass is left, the combined respiration is from both soil and plants, and the values of ER can be expected to be larger than where biomass is reduced. With an increase in temperature, affecting both microbial activity and plant enzyme activity, the respiration is expected to increase [46,47]. The GEP represents the total amount of $CO_2$ that the mesocosm takes up, and the potential uptake depending on the amount of biomass left in a mesocosm. The biomass left in the system will respire and absorb $CO_2$ corresponding to its leaf area and enzymatic functioning. The temperature as affecting enzymatic activity will likewise influence the GEP. The Figure 6 represents the development of ER and GEP in the two soils after the biomass cut and during the temperature campaign.

The Auzeville-Tolosane soil exhibits differences in the ER responses (Figure 6). In the CT climate, the response according to partial harvest is most visible in the grain legume rotation where also the response to daily increments of temperature is visible (Figure 7). Having grain legumes in the rotation seems to have no visible impact on the ER across the other climates. The GEP measurements are influenced by the partial harvest in all climates and both rotations. The GL1-rotation in the A climate demonstrates simultaneously the GEP dependency on biomass cover and temperature as GEP increases in all degrees of harvest at all days of measurement progressively with the increase in temperature. In the GL0-rotation there seems to be no effect of the day i.e., the temperature increases, indicating a smaller response to temperature in the rotation without than in the one with grain legumes.

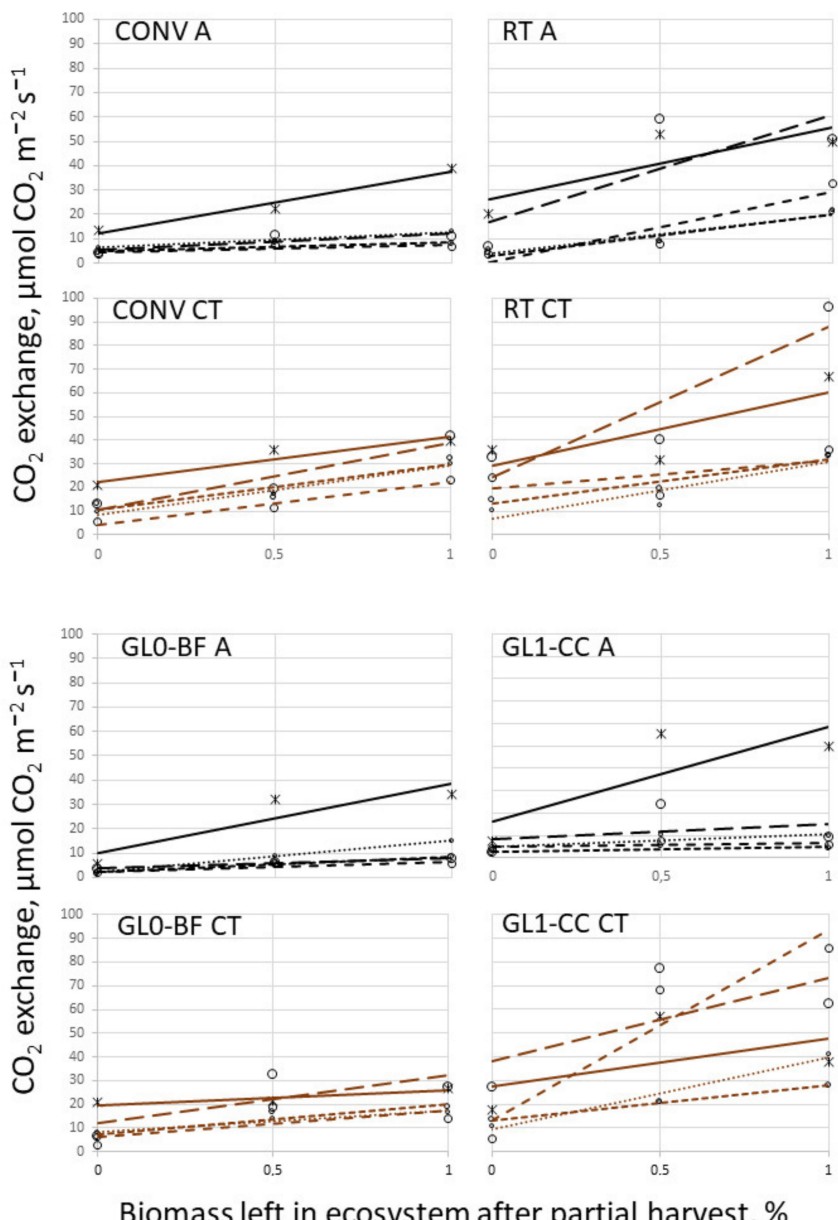

**Figure 6.** The ecosystem respiration (ER) in response to 0, 50, or 100% of the aboveground plant biomass left in the three columns of similar origin in each climate treatment. Mons (**top**) and Auzeville-Tolosane (**bottom**). Each trend line represents a measuring day that corresponds to an increase in temperature. The lines indicate timing of measurements: the smaller the dots or dashes the earlier the measurements, and the more solid the later the measurements. For further information about the treatment abbreviations, see Table 1 and Figure 2.

The present study shows interactions between the partial harvest and the climates in the response to temperature and days passing. As temperature increases, days pass, any biomass left in the mesocosm will grow, with the severely cut biomass growing the most (see Figure 8). A constantly increasing growth rate for all the mesocosms in a climate treatment would result in progressively increasing GEP as visible for GL1 in climate A (trendlines corresponding to days follow each other in an orderly manner). However, in the other climates (especially in GL1 CT) there seems to be a slowdown of growth increase when the temperature increases (the trendlines indicate increases at first and then decreases at the later days). In the climate treatments, especially with increased temperature, GL1 CT, the GEP changes with the days to become lower than the initial GEP

of the first days' measurement. There is no discernible difference due to timing in GL0 A or CT. The incorporation of grain legumes in the rotation results in a more variable response of GEP to increase in temperature compared to no grain legumes in the rotation where the GEP response is restricted mainly to the percent of biomass left in the mesocosm but it is throughout a little smaller than the GL1-rotation GEPs.

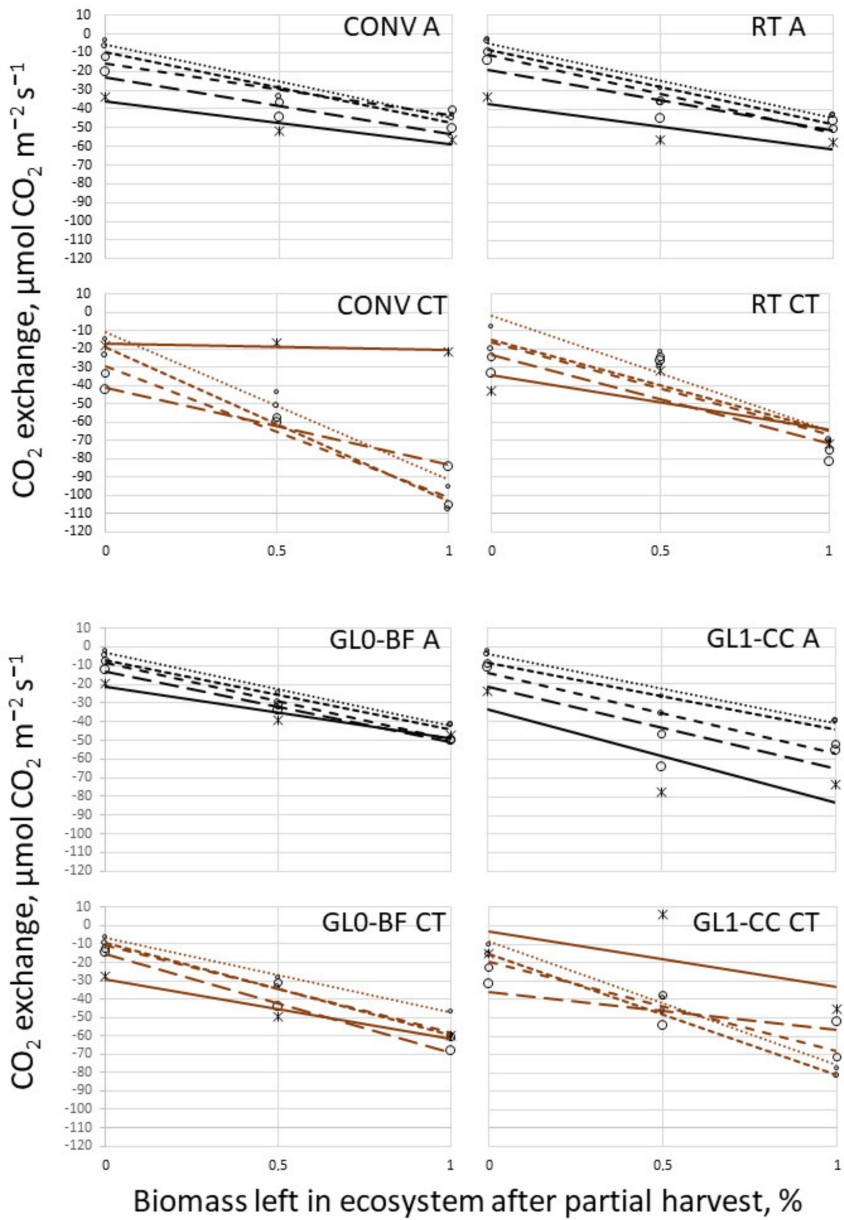

**Figure 7.** The gross ecosystem production (GEP) in response to 0, 50, or 100 % of the aboveground plant biomass left in the three columns of similar origin in each climate treatment. Mons (**top**) and Auzeville-Tolosane (**bottom**). Each trend line represents a measuring day that corresponds to an increase in temperature. The lines indicate timing of measurements: the smaller dots or dashes the earlier the measurements, and the more solid the later the measurements. For further information about the treatment abbreviations, see Table 1 and Figure 2.

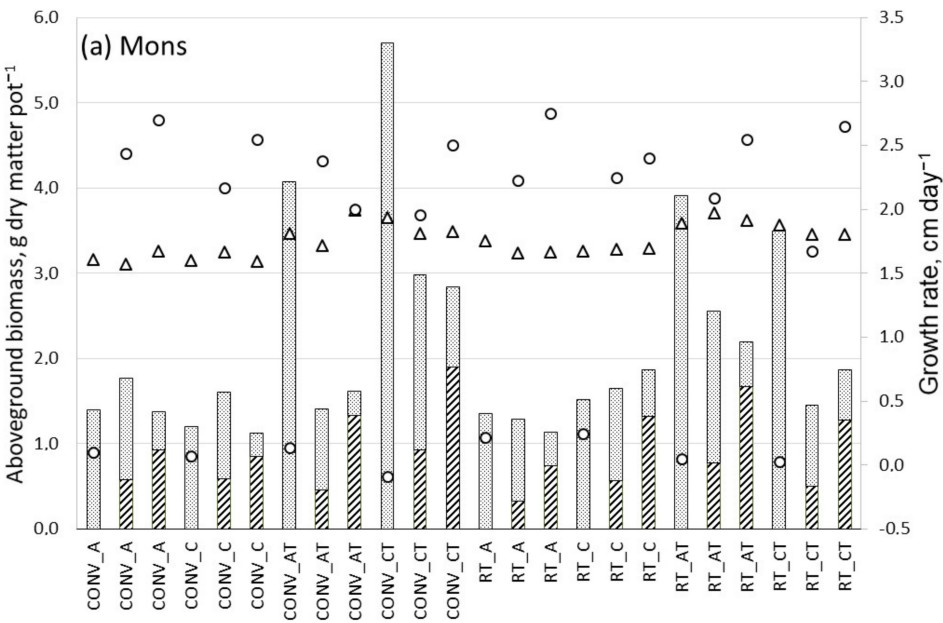

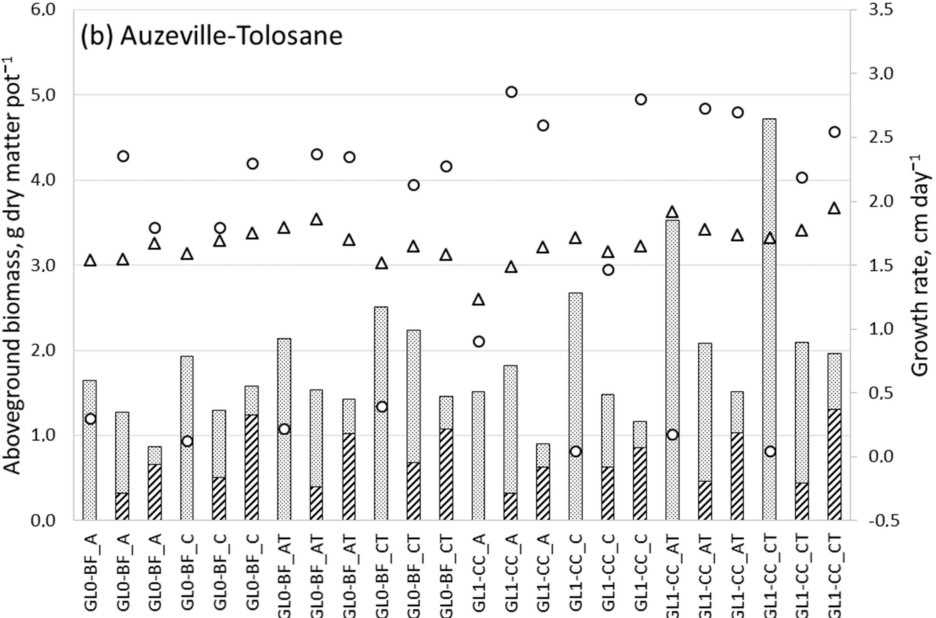

**Figure 8.** The biomass from the individual soil columns from Mons (**a**) and Auzeville-Tolosane (**b**). The bars reflect if the biomass has been harvested or not. The obliquely hatched is the first round of harvest and the dotted bars indicate the final harvest. Triangles indicate the barley growth rate in cm day$^{-1}$ at the time from sowing to partial harvest, and the circles indicate the growth rate from partial harvest to final harvest.

The ER from the Mons soils with different tillage strategies is in the climate A and CT measurements higher from the RT management than from the CONV management. The pattern of GEP for the Mons soil indicates that the effect of temperature (AT vs. A) is an accentuation of the difference between the degrees of biomass harvested. There is little difference of GEP between the CONV and RT managements in climate A; and the GEP increases with day (~temperature) as could be expected with increase in temperature and growth of plants. The CT GEP in Mons CONV and in Auzeville-Tolosane GL1-CC seem to reach a plateau or be skewed out of optimum on the latest days of measuring; a

development that seems to be on its way but has not yet happened in the Mons RT and Auzeville-Tolosane GL0-BF. The ecosystem responses to the increments in temperature seems to be related to the management history of the soils.

### 3.4. Barley Total Aboveground Dry Matter Production

The aboveground biomass represents the processes of carbon assimilation; the availability of substrate, the biochemical performance of the plant enzymes, the soil provision of nutrients and water, temperatures; the better the conditions for the plant to assimilate carbon, the more biomass would be produced. The aboveground biomass forms the basis for the production of yield later, and since maturity is out of scope for a spring settings comparison, we use the aboveground biomass produced by partial and final harvests within the 34 days of growth for each soil sample (Figure 8).

There was statistically significant effect of interactions between temperature, carbon-enrichment, and soil treatment for Mons soil columns ($p = 0.0062$). Each of these factors contributes to the regulation of plant growth by potentially limiting enzymatic processes or by providing nutrients, anchorage, and water.

The interaction between the factors represent the many possible processes to be influenced by changes and according to previous work on changes of single and combined factors' influence on plant growth, interactions are expected [48]. However, in the Auzeville-Tolosane soil columns, only the temperature level induced a significant effect on the total biomass produced; where higher temperature had a negative effect ($p = 0.0096$). Thus, neither changes in CO2-availability nor the addition of grain legumes to the rotation had an impact on the biomass production in these soils. Like shown in the present study, the fertilizing effect of increased atmospheric CO2 may be smaller than expected in the long term [49]. Grain legumes included in the rotation is considered beneficial for subsequent crops [14,50], and according to Plaza-Bonilla et al. [35] grain yield of these soils showed significant influence from preceding rotation. Nevertheless, in this short-term study no significant difference was found in the aboveground biomass production as a result of rotation history.

The barley DM output in this experiment is suitable to indicate that should farmers decide to reduce tillage or include grain legumes in the crop rotation, the biomass output is not expected to decrease or increase significantly in the initial stages of growth even under changed climatic conditions. This is in accordance with the literature where changes in yields from reducing tillage reportedly are as varied as interannual variations [51], but literature tend to be more favorable of legumes inclusion's impact on yield [50] than what is discernible here. With an undisturbed soil profile the intact soil-core microcosm set-up used in the present study is believed to be an improvement over more traditional pot experimentations. Nevertheless, predictors of field effects representing the complexity of natural systems and the multitude of factors in a field situation are not included and further farming cropping system elaboration of the present study should be conducted with care.

### 3.5. Nitrous Oxide

Nitrous oxide production in soil depends on soil heterogeneity, soil properties, and carbon content and hence nitrous oxide fluxes exhibit normally a large spatio-temporal variation [29,52,53]. The legume plant residues in the soil with the grain legume rotation increase the heterogeneity by adding additional organic nitrogen sources to the soil. Thus, the rotation and the climate settings may influence the nitrous oxide fluxes. The set-up of the experiment and the sampling method does not have the benefits of automatized chambers described by Peyrard et al. [54], however, the uniformity of conditions in the climate chambers offer the benefit of comparability regarding the different climate settings. In the present study there is no statistically significant differences to report presumably due to the heterogeneity of the soils linked to the limited number of replications in this experiment to represent the field-scale variability (Figure 9).

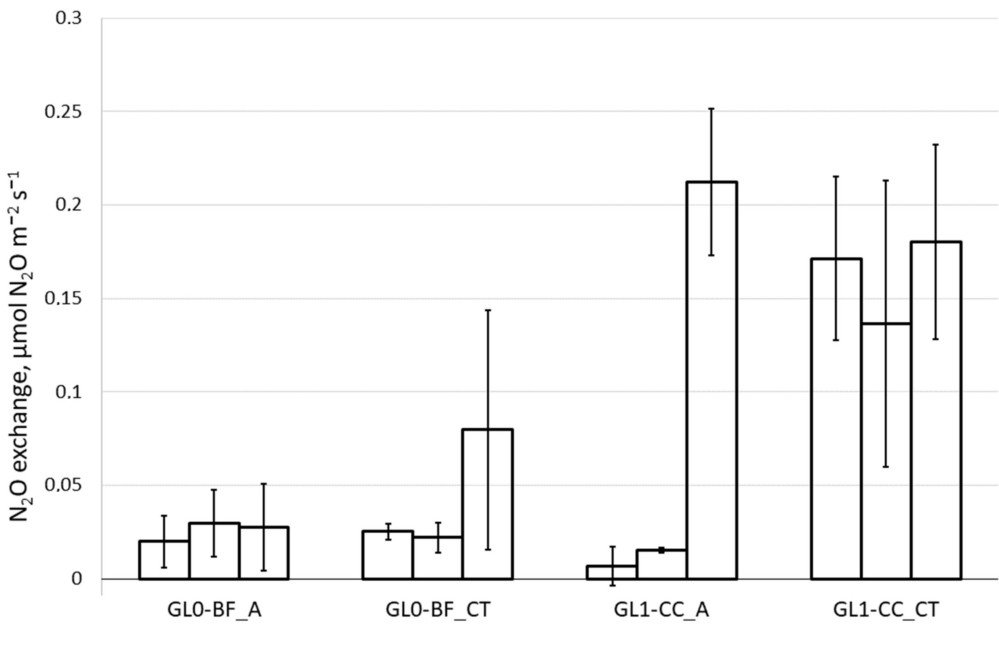

**Figure 9.** Nitrous oxide ($N_2O$) flux measurements from the Auzeville-Tolosane soils with the rotation treatments (i) no grain legumes + bare fallow (GL0-BF) and (ii) grain legume + cover crops (GL1-CC) placed in climates A and CT (see Figure 2), with standard errors representing variability of the three sampling dates. The three columns show the individual soil mesocom replication in the climate treatments.

The cover crops with legumes included in the rotation could result in a higher $N_2O$ emission as shown by Sanz-Cobena et al. [55] with higher emissions from soils including vetch as cover crop. In the present study, both a grain legume and a vetch-oat cover crop were included in the rotation, but the impact of the legumes was not displayed in the ambient climate treatment A (Figure 9). This is in line with the literature on $N_2O$ emissions from legume rotations and background control emissions [13]. However, the CT treatment with both elevated temperature and elevated $CO_2$ was different from the other three combinations indicating that in future warmer climate an increase in nitrous oxides from soils grown with grain legumes can be expected. This aligns with the findings of e.g., Kamp et al. [56].

*3.6. Climate Chambers vs. Field Conditions*

We think our setup works and that similar experimental system can be designed to improve understanding of agricultural practices under varied controlled environmental conditions. However, it is important to underline that the results from the climate chambers are not directly transferable to field conditions [57,58]. Differences found in such set-up can be attributed to e.g., differentiated managements and different settings of environmental conditions for comparisons and importantly increased mechanistic understanding. The findings from especially the ER, NEE, and GEP of the Mons soils and from the bare soil respiration of the Auzeville-Tolosane soil describe the need to test future climate conditions and ecosystem responses combining growing factors as responses of single factors are typically not transferable to treatments with factor combinations. We consider the current climate chamber approach as complementary to field work with approaches such as FACE and open top chambers, each having their own benefits and drawbacks. For the current climate chamber work, control, and reproducibility are of high value vs. work in natural environment with pronounced dynamics in soil conditions. Hence, for the climate chamber approach the option of connecting agricultural practice tightly with environmental conditions and plant growth is appreciated.

## 4. Conclusions

In this study, where long-term agricultural field-trials are combined with environmental treatments in climate chambers and plant growth, we propose a method to relate the history of the soil with the future of the climate. Applying and adjusting the method over time will direct its use.

The ecosystems' gas exchange patterns of the soil sample mesocosms reflect interactions between the agricultural management strategies and the climate treatments. The ecosystem respiration is higher or more diverse in reduced tillage treatment and grain legume rotation compared to conventional plowing and no grain legumes, respectively. The gross ecosystem production was higher or more diverse in the soil samples with conventional plowing and grain legume in rotation compared to the soil samples with reduced tillage and no grain legumes, respectively. The nitrous oxides exchanges from the soil samples are highly variable, especially in the elevated temperature and elevated $CO_2$ treatment with an influence on the grain legume rotation. The biomass production in the initial period of growth was not significantly influenced by the management or environmental conditions. The influence from climate treatments caused only enough differences in respiration and ecosystem production from the soil sample mesocosms to blur a decisive verdict of plowing being superior to reduced tillage or legume rotation to no legumes.

**Author Contributions:** Conceptualization, E.M.Ø.H., H.H.-N., E.J., and T.N.M.; methodology, E.M.Ø.H., H.H.-N., P.A., and T.N.M.; formal analysis, E.M.Ø.H., E.J., P.A., and T.N.M.; writing—original draft preparation, E.M.Ø.H.; writing—review and editing, E.M.Ø.H., H.H.-N., E.J., T.N.M., and P.A.; visualization, E.M.Ø.H.; supervision, T.N.M. and H.H.-N. All authors have read and agreed to the published version of the manuscript.

**Funding:** This research was funded by Joint Programming Initiative on Agriculture, Food Security and Climate Change (FACCE-JPI); the FACCE-ERA-NET+ project Climate–CAFÉ.

**Institutional Review Board Statement:** Not applicable.

**Informed Consent Statement:** Not applicable.

**Data Availability Statement:** The data presented in this study are available on request from the corresponding author.

**Acknowledgments:** The authors would like to thank for the technical assistance of Frédéric Bornet, Eric Grehan, Poul T. Sørensen, Mette Flodgaard, Didier Raffaillac, Eric Lecloux, and the logistic aid of Daniel Plaza-Bonilla. A special thanks to Bruno Mary for provision of intact soil columns to the project.

**Conflicts of Interest:** The authors declare no conflict of interest.

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
