# Peer review of "The Influence of Grain Legume and Tillage Strategies on CO2 and N2O Gas Exchange under Varied Environmental Conditions"

_agriculture, doi:10.3390/agriculture11050464_

Round 1
Reviewer 1 Report
The article "The influence of grain legume and tillage strategies on CO2 and
N2O gas exchange under varied environmental conditions" looks to explore greenhouse gas emissions (specifically nitrous oxide and carbon dioxide) from agricultural fields under different management practices. As such this paper is one of many that have examined the impact of agricultural practise on GHGs, providing a useful data point but unlikely to substantively change opinions or perspectives.
The paper overall is well written, although there will need to be some clean up in terms of precise use of english grammar. Having said this, the introduction and the abstract both focus on agricultural soils and soil carbon, which are not really addressed in this work. There is no reporting on soil carbon or other soil parameters, so starting with this approach is a bit misleading relative to the actual paper aims. Speaking of which, I confess the aims of the paper and the specific work performed is not clear to me after reading the introduction.
After reading through the methods I have serious concerns about the reported outcomes from this experiment. Many of the outcomes seem to be the result of an over analyzed set of data and insufficiently aware of previous work on these themes. Both in terms of reported results AND in terms of known methodological issues.
Main concerns listed below:
Soil monoliths will have substantial necromass from root and hyphal cutting during collection, and often have greater than normal respiration for months after. How have the authors addressed this concern?
The sampling approach can lead to substantial compaction of the soil column. Did the authors measure soil column height before and after collection? Also, it seems that the wick system employed will reverse standard soil moisture profiles, which is likely to change root behaviour and trace gas fluxes.
The soils are not particularly similar across sites, particularly in terms of silt/sand content. As I read through this I strongly object to the idea that soil management effects can be effectively separated from soil sampling location. I think that these conclusions need to be revisited with an eye towards site-dependence leading most of the observable impacts. Suggesting that soil management can be compared across sites is a non-starter for me. Perhaps within site variations can be ascribed, but I wouldn't want to compare reduced tillage from Mons to the legume+cover crop from Auzeville-Tolosane.
The length of the study (23 days post emergence then harvest?) is insufficient to really draw any conclusions regarding long term, in-field effects. I am uncertain why this time frame was chosen- is this intended to reflect grazing pastures? If so, why wasn't this discussed earlier in the text? I gather from some text that this reflects spring growth, but plant success is not really discussed in the paper, while GHGs are, so this does not seem to be a sufficient reason for the limited growing period.
In terms of the trace gas fluxes it has been established for a long time that internal fans can cause substantially enhanced trace gas fluxes. How have the authors addressed this concern? Also- while I appreciate the effort required for the trace gas analyses, individual timepoint CO2 analyses can be inaccurate in reflecting diurnal NEE/RE cycles. This is not sufficiently addressed. Furthermore, Los Gatos flux measurements often are skewed in the first few moments after chamber emplacement due to pressure variations- how were these changes taken into account? Normally there is a delay between emplacement and the start of the linear response. What was this time frame, and/or how did you determine linearity? Finally, we know that trace gas fluxes change dramatically throughout the season, so the rationale for the very limited sampling period is even more unclear.
It is also clear that some plant-climate adaptation will occur between now and the time at which the chosen temperature modifications will occur. This needs to be addressed in the discussion, since plant acclimation/adaptation strategies may ameliorate some/much of the differences observed in these types of studies.
Table 1 is not clear to me, specifically the three columns on the right hand side- are these all only about Mons soils? OK- having read through the caption 3 times I now understand, but this could be easier!
I am assuming that the error bars in Fig 5 are standard error.
Unclear of the value of the trendlines in Fig 6, these should probably be box and whisker plots, or something similar.
Author Response
Please, see attachment

Reviewer 2 Report
The idea behind this paper is fascinating and updated. I would like to congratulate the authors for their efforts in this manuscript. The topic fits with the journal and will be interesting for the readers. Nonetheless, some changes are required to ensure that the paper reaches the expected quality of this journal. Following, I include some comments aimed to improve the paper:
- In the abstract, I would suggest adding some of their results in numbers to highlight the major encountered differences or the most relevant findings.
- Given the importance of soil in the current efforts of EU and their calls and the relevance of soil in the paper, I propose adding the word soil as a keyword, for example, soil respiration (but authros can select another option).
- In the introduction, the authors have a paragraph in which they detail the aim of the paper. In the abstract, the authors have to highlight the novelty of their contribution compared with the previous studies.
- In section 2, if possible, a picture of the experiment have to be added.
- Subsection 2.7 must be improved. Authors have to detail the statistical analysis performed (for example, ANOVAs), the selected methods for generate the multiple ranges, etc…
- Section 3 must be divided into two sections. The first one (results) describes the results of their experiment and compares the differences between soils. Then in the second section (discussion), authors must evaluate their findings, comparing them with the results of outer authors. Finally, they have to discuss the impact of their findings related to the possible increase of CO2 and to the current tendency of recovering the conservation agriculture (which implies reduced tillage or no-tillage).
- In the graphics included in Figure 4, authors must include a legend detailing the information included in the caption (caption have to be reduced).
- For Figure 5, the authors have to add the letters in the graphic of Mons.
- In conclusions, authors must add a new paragraph describing the future work linked to their results.
Reviewer 3 Report
The research results presented in the manuscript seem to be interesting. Unfortunately, these are the results of a laboratory experiment, not a field experiment. Of course, it is difficult to imagine controlling the temperature and CO2 content in the air under field conditions, but (although the research results do not raise any doubts), they are a kind of reflection of a certain artificial experimental system. Nevertheless, these are valuable data allowing the prediction of changes in the intensity of natural phenomena under changing climatic conditions. In such cases, I always wonder to what extent the transfer of soil from the field to the vessels changes the biotope and, consequently, the intensity of microbiological processes. One might wonder whether the number of measurements of the amount of CO2 and N2O released from the soil is sufficient for this type of research. Should the defined hypothesis and research goal not be implemented as a result of continuous measurements or at least performed more frequently than assumed in the experiment (?) The duration of the experiment can be considered in the same context - it in no way relates to the duration of the vegetation period. Taking into account the time of natural soil compaction after its transfer, the period of the experiment could be much longer (especially with the adopted purpose of the work). The selection of statistical analysis methods to assess differences in GHG emissions is questionable. Literature data indicate that this phenomenon does not have the character of a normal distribution, but with the number of measurements adopted in the experiment, the use of non-parametric tests is impossible. So some statistical method had to be used, although in this case I don't know if descriptive statistics are better. However, since the research is original and, in a sense, pioneering, one may conditionally accept the adopted solution.
Round 2
Reviewer 1 Report
The authors have been much more clear about their intentions for this research in the introduction and abstract, allowing me to better understand that this paper presents a description of a methodological approach which the authors feel may be useful.
This addresses several of my concerns from the first review but elevates two others. Particularly:
1) There is effectively no recognition of other approaches that are used to address the same types of climate/landscape interactions. I am thinking of FACE experiments, as well as open chamber heating approaches, among others. How does this experiment improve upon, or deliver different outcomes relative to these other, published approaches? In order to make the case that this method is useful it would be good to have this discussion at greater depth in this paper.
2) I still feel that many of the claims in the results and discussion are over-weighted relative to the methodologies used. These are not truly generalizable outcomes, and they tend to be reported as such. I would prefer that in the results and discussion that it be recognized that these outcomes are reflective of this study, and this study alone. I wouldn't want people to read this article and take away that land management techniques had a particular implication for respiration and/or nitrous oxide emissions since this experiment is insufficient in time and soil impact consideration to allow such certainty.
